True’s beaked whale (Mesoplodon mirus) in Macaronesia

Aguilar de Soto Natacha naguilar@ull.es 1 2 3
Martín Vidal 4
Silva Monica 5 6
Edler Roland 7
Reyes Cristel 2 8
Carrillo Manuel 9
Schiavi Agustina 2
Morales Talia 2 3
García-Ovide Belen 2 3
Sanchez-Mora Anna 2 3
Garcia-Tavero Nerea 2 3
Steiner Lisa 10
Scheer Michael 11
Gockel Roland 11
Walker Dylan 12
Villa Enrico 13
Szlama Petra 13
Eriksson Ida K. 14
Tejedor Marisa 4
Perez-Gil Monica 4 15
Quaresma João 16
Bachara Wojtek 17
Carroll Emma 18
1 Centre for Research into Ecological and Environmental Modelling (CREEM), University of St. Andrews , St Andrews, Scotland , United Kingdom
2 BIOECOMAC. Department of Animal Biology, Edaphology and Geology, University of La Laguna , Tenerife , Canary Islands , Spain
3 Grupo de Investigación en Conservación (GIC) , Tenerife , Canary Islands , Spain
4 Society for the Study of Cetaceans in the Canary Islands Archipelago (SECAC) , Lanzarote , Canary Islands , Spain
5 Institute of Marine Research (IMAR), University of the Azores, Marine and Environmental Sciences Centre (MARE) , Faial , Azores , Portugal
6 Department of Biology, Woods Hole Oceanographic Institution , MA , United States
7 Zoo Duisburg , Duisburg , Germany
8 Sea Mammal Research Unit (SMRU), University of St. Andrews , St Andrews , Scotland , United Kingdom
9 Canarias Conservacion , Tenerife , Canary Islands , Spain
10 Whale Watch Azores , Faial , Azores , Portugal
11 M.E.E.R. La Gomera , Berlin , Germany
12 World Cetacean Alliance , Brighton , England , United Kingdom
13 Cetacean Watching (CW Azores) , Pico , Azores , Portugal
14 Futurismo , Ponta Delgada , Azores , Portugal
15 CEAMAR, Cetaceans and Marine Research Institute of the Canary Islands , Lanzarote , Spain
16 Espaço Talassa , Lajes do Pico , Azores , Portugal
17 Legionowo , Poland
18 Scottish Ocean Institute, University of St. Andrews , St Andrews , Scotland , United Kingdom
Reimer James
Electronic publication date: 2017 Mar 7
Publication date: 2017
Volume: 5
Electronic Location ID: e3059
Received 2016 Feb 25; Accepted 2017 Feb 3
Copyright: ©2017 Aguilar de Soto et al.
Copyright year: 2017
Copyright holder: Aguilar de Soto et al.
License: This is an open access article distributed under the terms of the Creative Commons Attribution License, which permits unrestricted use, distribution, reproduction and adaptation in any medium and for any purpose provided that it is properly attributed. For attribution, the original author(s), title, publication source (PeerJ) and either DOI or URL of the article must be cited.
License URL: https://creativecommons.org/licenses/by/4.0/

Keywords: Ziphiidae, Cytochrome b, mtDNA, Genetics, Colouration patterns, Phenotype, North Atlantic, Cetacean distribution, Canary Islands, Azores

Funding: Canary Islands Government and the Spanish Ministry MAPAMA Fundación Biodiversidad-MAPAMA (2015) ONR (2010–2014 and 2016) FCT and FRCT TRACE-PTDC/MAR/74071/2006 MAPCET-M2.1.2/F/012/2011 FEDER, COMPETE, QREN European Social Fund, and Proconvergencia Açores/EU Program POPH, QREN European Social Fund and the Portuguese Ministry for Science and Education Newton International Fellowship from the Royal Society of London EU-FP7 Marie Curie project SOUNDMAR EU-Horizon 2020 Marie Slodowska Curie projects ECOSOUND and “Behaviour-Connect” Stranding data collection in the Canary Islands is performed by the Canary Islands Cetacean Stranding Network, funded by the Canary Islands Government and the Spanish Ministry MAPAMA. Long-term monitoring of beaked whales in El Hierro was funded by Fundación Biodiversidad-MAPAMA (2015) and ONR (2010–2014 and 2016). The Cetacean and Seabird Sighting Network of te Canary Islands (CETAVIST) was supported to University of La Laguna and GIC by Fundación Biodiversidad-MAPAMA within the project Canarias con la Mar. Strandings data collection in Azores was funded by FCT and FRCT, through TRACE-PTDC/MAR/74071/2006 and MAPCET-M2.1.2/F/012/2011 (FEDER, COMPETE, QREN European Social Fund, and Proconvergencia Açores/EU Program). MAS is supported by an FCT-Investigator contract (funded by POPH, QREN European Social Fund and the Portuguese Ministry for Science and Education). ELC was supported for the analysis by a Newton International Fellowship from the Royal Society of London and during writting by the EU-FP7 Marie Curie project “Behaviour-Connect”. NAS was funded during data collection of this work by the EU-FP7 Marie Curie project SOUNDMAR and during writting by project ECOSOUND within the Horizon 2020 EU Marie Slodowska Curie program. The funders had no role in study design, data collection and analysis, decision to publish, or preparation of the manuscript.

==============================
The True’s beaked whale (Mesoplodon mirus, True 1913) is a poorly known member of the Ziphiidae family. Its distribution in the northern hemisphere is thought to be restricted to the temperate or warm temperate waters of the North Atlantic, while a few stranding records from the southern hemisphere suggest a wider and antitropical distribution, extending to waters from the Atlantic coast of Brazil to South Africa, Mozambique, Australia and the Tasman Sea coast of New Zealand. This paper (i) reports the first molecular confirmation of the occurrence of the True’s beaked whale at the southern limit of its distribution recorded in the northeast Atlantic: the Azores and Canary Islands (macaronesian ecoregion); (ii) describes a new colouration for this species using evidence from a whale with molecular species confirmation; and (iii) contributes to the sparse worldwide database of live sightings, including the first underwater video recording of this species and close images of a calf. Species identification was confirmed in two cases using mitochondrial DNA control region and cytochrome b gene markers: a subadult male True’s beaked whale that stranded in El Hierro, Canary Islands, in November 2012, and a subadult male found floating dead near Faial, the Azores, in July 2004. The whale that stranded in the Canary Islands had a clearly delimited white area on its head, extending posteriorly from the tip of the beak to cover the blowhole dorsally and the gular grooves ventrally. This colouration contrasts with previous descriptions for the species and it may be rare, but it exemplifies the variability of the colouration of True’s beaked whales in the North Atlantic, further confirmed here by live sightings data. The recording of several observations of this species in deep but relatively coastal waters off the Azores and the Canary Islands suggests that these archipelagos may be unique locations to study the behaviour of the enigmatic True’s beaked whale.

Introduction

Studies on animal distribution rely on the correct identification of the focal species during surveys. This can be challenging for marine mammals that are present at the sea surface for short time periods, particularly when they share colouration patterns and morphology with closely related species. These challenges are exemplified by the family Ziphiidae, which contains 22 species of beaked whales. Within ziphiids, species of the genus Mesoplodon have been proposed as some of the most poorly known of all the genera of large mammals (Jefferson, Pitman & Webber, 2015). Ziphiids dive to depth for long periods of time, with only short breathing intervals at the sea surface (Tyack et al., 2006; Aguilar de Soto et al., 2012). Furthermore, they show large intraspecific variability in colouration and interspecific similarities in general morphology, including colouration patterns (Mead, 2009; Jefferson, Pitman & Webber, 2015). Due to the inherent difficulties in identifying beaked whales to species level at sea, sightings of different ziphiid species are often pooled for analyses of survey data (e.g., Moore & Barlow, 2013). This results in a loss of precision in our knowledge about the distribution of individual species.

Beaked whales are broadly distributed in all oceans of the world. Six species of three genera can be found regularly in the North Atlantic: Cuvier’s beaked whales (∼5–7 m) and northern bottlenose whales (∼9–10 m) (Ziphius cavirostris and Hyperoodon ampullatus, respectively), and four species of the genus Mesoplodon (∼4.5–5.5 m): Blainville’s, Sowerby’s, Gervais’ and True’s beaked whales (M. densirostris, M. bidens, M. europaeus and M. mirus, respectively) (MacLeod et al., 2006). The large size and distinctive head morphology of Cuvier’s beaked whales and bottlenose whales facilitate their differentiation at sea. In addition, adult Cuvier’s beaked whales often show clear colour patches in the head, dorsum and other parts of the body, and this provides a further identification cue (examples at http://www.cetabase.info). Mesoplodonts are similar in size and often difficult to identify at sea to species level. The position of the teeth along the lower jaw is the most reliable cue to distinguish adult males at sea. Females and subadult males do not have erupted teeth but the location of alveoli in the lower jaw, diagnostic of species identity, can be uncovered during necropsy of stranded whales (Jefferson, Pitman & Webber, 2015). Beak and melon size and shape can be used as defining characteristics of the Mesoplodont species. For example, both Blainville’s and Sowerby’s beaked whales have relatively long beaks. However, these species can be distinguished in the field, as an arched lower jaw with protruding teeth often covered in barnacles is typical of the male Blainville’s beaked whales, while a bulky melon and thin long beak are characteristic of Sowerby’s beaked whales (Jefferson, Pitman & Webber, 2015; Mead, 1989). In addition, the distribution of these two species seems to only partially overlap, with the former preferring warmer waters than the latter (MacLeod et al., 2006). True’s and Gervais’ beaked whales can share a general grey colouration including a dark eye patch and a pale ventral area in some cases; both have shorter, mostly straight beaks. These two species are very difficult to distinguish at sea. The position of the teeth in the jaw of males provide the most definitive cue but teeth, even when present, are not always easy to observe at sea. A species-defining characteristic is the melon, which is bulbous and well defined in Sowerby’s beaked whales and also in True’s beaked whales, albeit less pronounced in the latter. In contrast, the melon of Gervais’ and Blainville’s beaked whales slopes gently towards the beak (Weir et al., 2004).

Live sightings of many beaked whale species are rare events and just a few have been made for True’s beaked whales. Only three live sightings have been reported in the peer-reviewed literature for the North Atlantic (Weir et al., 2004) and some (Tove, 1995) may be misidentified Gervais’ beaked whales. The distribution of True’s beaked whales was once thought to be restricted to temperate or warm-temperate waters of the North Atlantic. However, strandings of animals in South Africa (McCann & Talbot, 1964; Ross, 1969; Ross, 1984), Mozambique (Bachara & Gullan, 2016), Australia (Dixon & Frigo, 1994), and Brazil (Mead, 1989; De Souza et al., 2004), revised in MacLeod et al. (2006), as well as in New Zealand (Constantine et al., 2014), extended its known range to temperate waters of the Southern Indian and South Atlantic Oceans and the Tasman Sea. Thus, True’s beaked whales have an antitropical distribution that is unique among ziphiids (Mead, 1989; MacLeod et al., 2006). In the North Atlantic, the southernmost limit of the distribution of True’s beaked whales is Macaronesia. The macaronesian ecoregion contains the Azores, Madeira and Canary Islands archipelagos (Spalding et al., 2007). For highly migratory species, it has been proposed that the southern Cape Verde archipelago should also be included in the ecoregion (Brito Hernández, 2010). In the Azores, a 3.7 m long subadult male True’s beaked whale with unerupted teeth was found dead in 2004, drifting south of the Faial-Pico channel (Silva et al., 2014). In the Canary Islands, a 5 m long adult male True’s beaked whale stranded in 1984 on the island of Lanzarote (Vonk & Martin, 1988). Both animals were identified as True’s beaked whales by their general morphology. The species has neither been recorded in Madeira (Freitas et al., 2012) nor Cape Verde archipelagos (Hazevoet et al., 2010).

Molecular markers are highly useful to diagnose species among cetaceans (Ross et al., 2003) and often more definitive than morphological characteristics that can be difficult to distinguish at sea or when animals strand in a decomposed state (e.g., Dalebout et al., 2002; Constantine et al., 2014). This is particularly applicable to beaked whales (Dalebout et al., 2004; Thompson et al., 2013). This paper reports the first occurrence of True’s beaked whales in the Canary Islands and the Azores confirmed with molecular markers. Furthermore, the whale that stranded in the Canary Islands showed a colouration pattern that has not been previously described for this species. These findings are augmented with new live sighting data of True’s beaked whales off the Azores and the Canary Islands, suggesting that these archipelagos are potentially good areas to study the natural behaviour of this species. Sightings are supported with video and photographic material including the first underwater recording of True’s beaked whales in the wild and close-up images of a calf.

Methods

Strandings and genetic analysis

A 3.9 m long beaked whale stranded at Timijiraque, El Hierro, the Canary Islands, on 30 November 2012 (Fig. 1). Observers at the beach reported that the animal might have live-stranded. The whale was identified as an immature male True’s beaked whale by its external morphology. No teeth had erupted nor were any present in the lower jaw. A 3.7 m long subadult male was found drifting south of the Faial-Pico channel, the Azores, on 11 July 2004 (Silva et al., 2014) (Fig. 2). It was identified to species-level by its general morphology and two small non-erupted teeth in the tip of the lower jaw.

Figure 1 True’s beaked whale stranded at El Hierro (Canary Islands) in 2012 showing a head colouration not described previously for this species (report 6 in Table 1).

Photos: Baudilio Quintero.

Figure 2 True’s beaked whale found drifting south of Pico-Faial channel (the Azores) in 2004 (report 4 in Table 1).

Photo: Mónica Silva (MARES-IMAR. UA).

Skin samples were taken from both carcasses and preserved in 95% ethanol. Total genomic DNA was isolated using standard proteinase K digestion and phenol/chloroform methods (Sambrook, Fritsch & Maniatis, 1989) or a DNeasy kit (Qiagen). Sex was confirmed by amplification of the male-specific SRY gene, multiplexed with an amplification of the ZFY/ZFX region as a positive control (Aasen & Medrano, 1990; Gilson et al., 1998). In order to confirm species identification, we amplified regions of both the mitochondrial DNA (mtDNA) control region and cytochrome b gene using polymerase chain reaction (PCR). Approximately 300 bp of the mtDNA control region were amplified using primers M13dlp1.5 (Baker et al., 1998) and Dlp4-H (Dalebout et al., 2005) and approximately 200 bp of the cytochrome bgene using CYBMF-L and CYBMR-H primers following standard protocols (Dalebout, 2002). These short fragments were targeted because the tissue, and hence DNA, was degraded as samples were collected sometime after death. PCR products were purified for sequencing with AMPURE XP (Agilent) and sequenced with BigDye™ v3.1 Terminator Chemistry (Applied Biosystems) on an ABI 3130 XL. Resulting sequences were aligned against other beaked whale mtDNA sequences and edited by eye in Geneious v7 (http://www.geneious.com, Kearse et al., 2012) for sequence quality. Species identification was made using the DNA surveillance website, constructing a neighbour joining tree with the support of 1,000 bootstraps (Ross et al., 2003), and by comparing the target sequences with other beaked whale sequences available from GenBank using blast (Altschul et al., 1990; http://www.ncbi.nlm.nih.gov/genbank/) (Dalebout et al., 2004). A 3.5 m immature female True’s beaked whale was found stranded in Fuerteventura, Canary Islands, in 2004 (Fig. 3) and identified initially as Gervais’s beaked whale, but then re-classified as True’s due to the typical mouthline of the species and general morphology. This female stranded with fishing gear (long-line) entangled in the caudal peduncle. No tissue was preserved for molecular analysis.

Table 1 Reports of strandings (Strand.) and live sightings (Sight.) of True’s beaked whales in the Azores and the Canary Islands (CI) with information on location, date, number of whales (No) and certainty on the identification (“sure”, achieved using molecular genetic markers (G) or morphology (M), or “possible”).

Sightings classified as “possible” may be of True’s or Gervais‘ beaked whales. One sighting (#8 is probably of Gervais’ beaked whales). The entities that gathered the reports were: SECAC, Society for the Study of Cetaceans in the Canary Islands Archipelago; WW Azores, Whale watch Azores; UA, University of the Azores; ULL, University of La Laguna; CW, Azores whale watch; Master Mind educational program; Futurismo whale watch; Espaço Talassa whale watch.

Report	Location	Date (entity)	Lat, N	Lon, W	Depth	No	Behaviour	Certainty	
1. Strand.	Lanzarote, CI	23/03/1984 (SECAC)	28.984	13.5	–	1	Stranded	Sure (M)	
2. Sight.	Pico, Azores	07/09/1994 (WW Azores )	38.36	28.3767	1,200	3	Travelling	Possible (M)	
3. Strand.	Fuerteventura, CI	06/06/2004 (SECAC)	28.1336	14.24	–	1	Stranded	Sure (M)	
4. Strand.	Faial, Azores	11/07/2004 (UA)	38.47	28.64	–	1	Drifting dead	Sure (G)	
5. Sight.	Lanzarote, CI	29/09/2009 (SECAC)	28.8346	13.5915	1,100	4	Travelling	Sure (M)	
6. Strand	El Hierro, CI	30/11/2012 (ULL)	27.268	17.914	–	1	Stranded	Sure (G)	
7. Sight.	Pico, Azores	31/07/2010 (CW Azores)	38.3396	28.3573	1,300	3	Breaching	Sure (M)	
8. Sight.a	Pico, Azores	05/05/2013 (MasterMind)	38.28	28.341	1,600	3	Milling	Sure (M)	
9. Sight.	Offshore. CI	27/09/2013 (ULL)	28.31	14.99	2,500	2	Breaching	Probable M. europaeus (M)	
10. Sight. b	S. Miguel, Azores	22/05/2015 (Futurismo)	37.6386	25.5101	600	2	Travelling	Sure (M)	
11. Sight.	Offshore. CI	11/07/2015 (ULL)	28.4503	14.7048	2,500	2	Breaching	Possible (M)	
12. Sight.	Pico, Azores	10/08/2016 (EspaçoTalassa)	38.3242	28.3517	1,250	2	Travelling	Sure (M)	
Notes.

a Sighting with underwater video.

b Sighting with small calf.

Figure 3 True’s beaked whale stranded at Fuerteventura (Canary Islands) in 2004 (report 3 in Table 1).

Photo: Vidal Martín (SECAC).

Live sightings

Here we report data on seven live sightings of True’s beaked whales in Macaronesia (Table 1) plus an ambiguous sighting classified as probable Gervais’ beaked whale. Sightings were categorised as sure (n = 5) or possible (n = 2) True’s beaked whales by close inspection of the colouration and morphology of the whales in the photographs taken during surveys. Teeth were observed in only one individual. This, and the poor quality of the photographs in some of the sightings, made identification challenging. Photos of live sightings were sharpened and their contrast augmented with software packages Photoshop and GIMP.

Five live sightings (four classified as sure) were recorded in the Azores. One of these sightings (report 8 in Table 1) occurred during a field cruise of the educational program Master Mint (http://www.master-mint.de). A group of three beaked whales surfaced and milled near a drifting small inflatable boat for about 10 min, breathing every 9.7 s on average. This allowed the observers to film the animals underwater (Fig. 4 and Video S1) providing high-quality images for species identification. Shorter sightings were recorded in the Azores by whale watching companies (Table 1, Figs. 5–8).

Figure 4 True’s beaked whale observed off Pico showing a pale blaze on the melon (report 8 in Table 1; Video S1).

Photo: Roland Edler (Duisburg Zoo).

Figure 5 Possible True’s beaked whale observed off Pico (report 2 in Table 1).

Note the parallel lineal scars on the dorsum, suggestive of being made by close paired teeth. Photo: Lisa Steiner (Whale Watch Azores).

Figure 6 True’s beaked whale observed off Pico showing a pale blaze on the melon (report 7 in Table 1).

Photo: Petra Szlama (CW Azores).

Figure 7 True’s whale with calf observed south of São Miguel, Azores (report 10 in Table 1).

Photo: Ida Eriksson (Futurismo).

Figure 8 True’s beaked whale observed off Pico (report 12 in Table 1).

Photo João Quaresma (Espaço Talassa).

Three live sightings were also recorded in the Canary Islands. Only one was classified as sure: a group of four True’s beaked whales observed in 2009 for 8 min during a cetacean research cruise performed by the Society for the Study of Cetaceans in the Canary Islands Archipelago (SECAC, Fig. 9, report 5 in Table 1). The whales were estimated to measure 3.5–4.5 m in length; the largest whale did not present erupted teeth and was tentatively classified as an adult female. Two shorter sightings of True’s or Gervais’ beaked whales were impossible to identify with certainty to species level and were classified as possible True’s beaked whale and possible Gervais’ beaked whale. These sightings were recorded by the Cetacean and Seabird Sighting Network of the Canary Islands “CETAVIST” (http://www.aviste.me) (Figs. 10 and 11; reports 9 and 11 in Table 1). CETAVIST undertook 1,300 surveys onboard passenger ferries from December 2012 to October 2016; these surveys were performed by volunteer observers resulting in a heterogeneous observation effort in the different inter-island channels of the archipelago.

Figure 9 True’s beaked whale observed off Lanzarote (report 5 in Table 1).

Photo: Vidal Martín (SECAC).

Figure 10 Probable Gervais’ beaked observed at the Canary Islands (report 9 in Table 1).

The identification of this animal as True’s or Gervais’ is difficult because it is not clear if the white patch on the back (photo A) is an artefact of the light or a real colouration of the animal indicating a Gervais’ beaked whale. Photos: Cristel Reyes (ULL).

Figure 11 Possible True’s beaked whales observed in the Canary Islands (report 11 in Table 1).

Photos: Antonio Portales (Cetavist, ULL).

Ethics

Samples of two dead whales in the Canary Islands and the Azores were gathered for genetic analysis with authorization obtained from the Cabildo Insular of El Hierro (permit number 7021/12_dic_2012) and from the Government of the Azores (permit number 4/CN/2004, issued by the Environment Directorate of the Azores). The sightings data from SECAC were gathered under research permit number 659 125 MAOT/15202 from the Spanish Ministry of Agriculture and Environment (MAGRAMA). Sightings gathered opportunistically from regular ferry and permitted whale watching/educational boat operations did not require specific research ethics authorisations or government permits.

Results

Strandings and genetic analysis

Genetic sex identification confirmed that both the whale that stranded on El Hierro (Canary Islands) and the whale found drifting in the Faial-Pico channel (Azores) were males. Robust support placed both the El Hierro query sequence (mtDNA control region: 98% bootstrap support; cytochrome b gene: 94% bootstrap support) and the Azores’ query sequence (mtDNA control region: 97% bootstrap support; cytochrome b gene: 94% bootstrap support) in a species-specific clade with True’s beaked whale sequences using DNA surveillance.

Furthermore, both the mtDNA control region and cytochrome b sequences from both the El Hierro and Azores’ males closely matched GenBank sequences identified as True’s beaked whales (BLAST accessed May and November 2015, respectively). True’s beaked whale sequences from whales that had stranded on the Atlantic coast of the USA (accession numbers U70465.2 and AY579525.1) were the closest matches to both the El Hierro sample (U70465.2: 99% sequence identity, E-value 9e–153 and AY579525.1: 98% sequence identity, E-value 4e–151) and the Azores’ sample mtDNA control region sequences (U70465.2: 98% sequence identity, e-value 7e–152 and AY579525.1: 100% sequence identity, E-value 7e–147). The two top matches against the cytochrome b sequence for both samples were: (i) a True’s beaked whale that stranded on the Atlantic coast of the USA, accession number AY579551.1 (El Hierro sample: 95% sequence identity, E-value 3e–70; Azores’ sample: 99% sequence identity, E-value 3e–112); and (ii) a sequence from a True’s beaked whale that stranded in New Zealand, accession number KF435028.1 (El Hierro sample: 94% sequence identity, E-value 3e–61, Azores’ sample: 95% sequence identity, E-value 4e–81). The sequences generated in this study have been archived on GenBank (accession numbers Azores CytB: KX375801, KX375802; Azores Dlp: KX150446; El Hierro CytB: KX375803 and El Hierro Dlp KX150445).

Live sightings

Photographs and a video recording from a live sighting of True’s beaked whales from across Macaronesia revealed individuals with a diagonal pale blaze on the head. Data from report 8 (Table 1, Fig. 4 and Video S1) revealed that this blaze extends dorsally from behind the blowhole to the top of the melon and reaches ventrally to the eye and the start of the mouthline. No obvious size differences were observed among the individuals in the group and none of the three whales had erupted teeth. The same pale blaze colouration pattern on the head was present on other whales observed in different encounters off the Azores, including a female-calf pair where both whales showed the pale blaze on their heads (Fig. 7; report 10 in Table 1) and a breaching female or subadult male (Fig. 6; report 7 in Table 1). In contrast, this pale-coloured blaze was not evident on other individuals identified as possible True’s beaked whales in the same archipelago (Fig. 5; report 2 in Table 1).

A ‘white mask’ was observed only in the male subadult whale that stranded on El Hierro, the Canary Islands (Fig. 1 and report 6, Table 1). This white colouration covering the whole anterior part of the head, including the melon, beak and lower jaw, has not been previously described for this species. The live sightings in the Canary Islands (Figs. 9–11; reports 5, 9, 11 in Table 1) showed a grey colouration pattern in the melon and pale lower jaw consistent with previous descriptions of the species. The two sightings classified as “possible” True’s beaked whales in the Canary Islandswere both observed to breach repeatedly (Figs. 10 and 11). This behaviour has been observed previously in True’s beaked whales (Fig. 12) but was observed also in Gervais’ beaked whales in the Canary Islands (Fig. 13, Video S2). Given the morphological and behavioural similarities between these two species (Fig. 14), we cannot exclude that whales classified as possible True’s beaked whales in this paper were, in fact, Gervais’ beaked whales.

Figure 12 True’s beaked whale breaching at the Bay of Biscay.

Photos. Dylan Walker (WCA).

Figure 13 Gervais’ beaked whales observed south of La Gomera in the Canary Islands (Video S2).

Photos: Michael Scheer (MEER).

Figure 14 Schematic drawing showing differences between Gervais’ (A) and True’s (B) beaked whales. Note the more pronounced melon of True’s beaked whales and the lines in the dorsum of Gervais’ beaked whales.

These lines are not present in all individuals of Gervais’ beaked whales, but have never been observed in True’s beaked whales. The genital white patch shown here in Gervais’ may appear in True’s beaked whales also. Drawing by Vidal Martín (SECAC).

Discussion

New records of data-scarce species, such as True’s beaked whales, are highly valuable in increasing our knowledge about the morphology, behaviour and distribution of these species. True’s beaked whales in the North Atlantic are described as greyish in colouration. Some individuals show a dark eye mark and a dark blaze in the upper part of the body from behind the blowhole to past the dorsal fin; some animals may show a pale ventral colouration, sometimes extending to the lower jaw, while other animals may have a pale blaze on their melon (Weir et al., 2004; Jefferson, Pitman & Webber, 2015). Differing colour patterns have been found for individuals from the southern hemisphere. For example, a female stranded in South Africa showed a whitish dorsal colouration including the dorsal fin and extending to the tail peduncle (Ross, 1984).

The True’s beaked whale stranded in El Hierro (Canary Islands) had a clearly delimited white mask covering the anterior part of the head from the blowhole and the gular grooves to the rostrum (Fig. 1). This pigmentation pattern does not seem to be the result of a post mortem discolouration. In fact, beaked whale colouration tends to darken after stranding because of decomposition, a phenomenon that does not explain the striking white colouration pattern on the head of the whale stranded on El Hierro (A Van Helden & T Pusser, pers. comm., 2016), but might increase the contrast between the white head and the grey body after stranding. However, the animal might have stranded alive and was in a fresh state when the photographs were taken. This would suggest that the contrasting white cephalic patch observed in the specimen stranded in El Hierro was present in the living animal. It cannot be dismissed that this unique colour pattern might be an ontogenetic trait. However, this pattern was not observed on the calf nor on any of the other individuals reported in this paper, some of which were likely to be subadults as the whale stranded at El Hierro.

The white head colouration described for True’s beaked whales increases the probability of confusing this species with Cuvier’s beaked whales in sightings when the beak is not observed. Cuvier’s beaked whales often have a white colouration on their rostrum and frontal part of the head. Moreover, the female True’s beaked whale observed with a calf off the Azores (Fig. 7) had a neck and melon colouration very similar to one of the typical head and neck colour patterns shown by Cuvier’s beaked whales (e.g., http://www.cetabase.info, whale code ZCH15: http://bit.ly/1sbuzVA), underlining the possibility of misidentifying these two species when the beak is not observed.

In contrast with its mother, the calf in Fig. 7 shows a very distinctive head colouration with a white diagonal band observed in other live sightings of True’s beaked whales. The light colour blaze on the melon may be a common feature for True’s beaked whales inhabiting the North Atlantic: it has been observed off the Azores in whales of different age classes, from a calf to adult or subadult animals, and previously in the eastern North Atlantic (Weir et al., 2004). A similar light coloured head blaze has been observed in True’s beaked whales of both sexes and with different sizes in the western North Atlantic, albeit the pale colour disappears rapidly in stranded whales (T Pusser, pers. comm., 2016). This suggests that the pale blaze in the melon may pass unnoticed in strandings unless stranded animals are very fresh.

In contrast with the colouration patterns described above, some True’s beaked whales in the North Atlantic tend to be more uniformly grey, although they may have a small pale mark in the genital-anus area (http://vertebrates.si.edu/mammals/beaked_whales/pages/mmi/mmi_sp_pg7.htm) (Weir et al., 2004). A whale with this mark was photographed at the Bay of Biscay (Fig. 12) and identified as a True’s beaked whale thanks to the consistent location of white points at the tip of the lower jaw in several successive photographs, strongly suggesting that these points are the teeth of the whale. A white genital patch, similar to that described in Trues’s beaked whales, was observed in a live sighting and strandings of Gervais’ beaked whales in the Canary Islands (Fig. 13; unpublished data from the Canary Islands Stranding Network by V Martín & M Carrillo, 2000–2006, http://bit.ly/1SdueM0).

True’s and Gervais’ beaked whales in the North Atlantic have several similar morphological features that challenge their differentiation at sea: the relatively short beak, mostly straight mouthline, overall colouration and dark eye patches. Figure 14 shows a schematic view of the morphological differences between True’s and Gervais’ beaked whales, including a straighter mouthline in True’s than in Gervais’, slightly upwards edge of the mouthline in True’s; a pale band on the melon of True’s (not always present); and a distinctive pattern of pale/dark stripes (or a pale patch) frequent in the dorsum of Gervais’ beaked whales (also not always present). Also, a cue to differentiate these species is the pronounced and rounded melon of True’s beaked whales, contrasting with the relatively more flat-topped melon of Gervais’ beaked whales sloping gently towards the beak (Figs. 13 and 15). However, this may be difficult to judge from photos taken from different perspectives. When present and visible, scarring patterns can also be used to distinguish between species due to the different position of the erupted teeth in males. While True’s beaked whales show parallel and linear scars with small gaps between adjacent scars, Gervais’ whales are expected to have single linear scars (Weir et al., 2004). Parallel scars are visible in Fig. 5 (report 2 in Table 1), suggesting that these animals were indeed True’s beaked whales.

Figure 15 Gervais’s beaked whale observed off Tenerife (Canary Islands).

Note the head morphology of this whale in comparison with True’s beaked whales. Photo: Sergio Hanquet.

Variability in colour patterns is not surprising for ziphiids. These animals often undergo ontogenetic changes in colouration, and this trait can vary even among individuals of the same size and sex class (Mead, 2009). For this reason, molecular markers providing definite identification among ziphiid species (Dalebout et al., 2004; Thompson et al., 2013) should be used, where possible, to confirm identification obtained using morphological cues (Dalebout et al., 2002; Constantine et al., 2014). Intraspecific variability in colouration for many ziphiids, as well as similarities in general colouration patterns, size/morphology and behaviour among most Mesoplodon species, challenge taxonomic identification at sea. This may cause a bias when assessing the relative abundance of Mesoplodon species in the North Atlantic. Animals for which recognition is challenging will be often classified during surveys as unidentified beaked whales, while recognizable animals will be classified to species level. This will result in an apparent lower relative abundance of species difficult to recognize at sea.

Given the presence of True’s beaked whales in the Azores and the Canary Islands, it would be expected that this species also occurs in the area between both archipelagos, including Madeira. However, knowledge about the distribution of beaked whales in the eastern North Atlantic Ocean is limited by the relative scarcity of offshore cetacean surveys in Macaronesia. Strandings and sightings of True’s and possible True’s beaked whales in the Azores and Canary Islands occurred from March to November (Table 1). However, sample size is too low to infer any conclusion about seasonality in occurrence. The scarcity of sightings of True’s beaked whales may reflect a low abundance and/or a general preference for deep waters far from the slope where limited survey effort has been made. The latter is supported by the near lack of sightings of True’s beaked whales in relatively nearshore deep waters along the slope of the Canary Islands where other beaked whale species are found routinely. Seasonal surveys in coastal deep waters off the Canary Islands up to 1,800 m depth in the last decade have recorded only one sighting of True’s beaked whales, while Cuvier’s, Blainville’s and Gervais’ beaked whales are observed or strand year round (Aguilar de Soto, 2006; Martín, 2011 and unpublished data from the same authors from 2003 to 2016; Arranz et al., 2014). Most sightings of True’s beaked whales reported in this paper occurred at deeper depths (Table 1) although not far from the slope (Fig. 16), showing that True’s beaked whales visit deep waters near the coast in some areas. The sighting in shallower waters included a small calf (report 10 in Table 1). Preference of mother-calf pairs for coastal deep waters in oceanic archipelagos has been suggested for other species such as Blainville’s beaked whales (Claridge, 2013).

Figure 16 Worldwide known distribution of True’s beaked whales (A) and locations of the reports included in this paper (B,C).

The 1,000 m depth contour is marked as a thicker light blue line. (A) distribution of the species in the Atlantic, courtesy of the Digital Beaked Whale Atlas of GIS in Ecology, based on data from MacLeod et al. (2006) (http://www.gisinecology.com/Digital_Beaked_Whale_Atlas/Accessing_Data_From_The_DBWA.htm).

The relative abundance of live sightings of True’s beaked whales in deep coastal waters off the Azores, and to some extent off the Canary Islands, suggests that these archipelagos could be ideal areas to research True’s beaked whales in the wild. This is relevant because the identification of hot-spots where some species of beaked whales are found with reliability has provided most of our current knowledge about the natural behaviour of ziphiids (Hooker & Baird, 1999; Tyack et al., 2006; Baird et al., 2006; Minamikawa, Iwasaki & Kishiro, 2007; Aguilar de Soto et al., 2012; Claridge, 2013).

The disjointed global distribution of True’s beaked whales has led some authors to suggest that there may be some degree of genetic isolation between the populations of the southern and northern hemispheres (MacLeod et al., 2006; Dalebout et al., 2007). These authors propose that more research is required in order to assess if the northern and southern hemisphere populations might represent different species. The precedents for this include cetacean species that were thought to have an anti-tropical distribution and were finally separated as different species: Hector’s and Andrews’ beaked whales. M. hectori and M. bowdoini, were separated from M. perrini and M. carlhubbsi, respectively. Furthermore, right whales (Eubalaena spp.) were separated into three species (Rosenbaum et al., 2000): the southern right whale (E. australis), with circumpolar distribution in the southern hemisphere, and the North Atlantic (E. glacialis) and North Pacific (E. japonica) right whales. The results of the genetic analysis shown here suggest a potential genetic structure with a phylogeographic pattern for True’s beaked whales, as the sequences from the Canary Islands and Azores’ matched most closely those True’s sequences on GenBank from the North Atlantic. However, more data are required to test this hypothesis. Given adequate sampling, a global analysis of connectivity would provide useful in understanding gene flow among the seemingly disparate areas of distribution of True’s beaked whales worldwide.

Supplemental Information

Video S1 Underwater video of True’ beaked whales recorded off the Azores by R Edler within the Master Mint program (report 8 Table 1)

Click here for additional data file.

Video S2 Gervais’ beaked whales in a group of four whales breaching repetitively in the Canary Islands, recorded by Roland Gockel (MEER)

Click here for additional data file.

Thanks to the Insular Government (Cabildo) of El Hierro, to the Government of the Canary Islands and to the Spanish Ministry of Environment MAPAMA for providing permit to access to the True’s beaked whale stranded at El Hierro. Thanks also to the ferry companies Armas, Fred Olsen and Trasmediterránea for embarking observers of the CETAVIST sighting net, and thanks to the enthusiastic work of these volunteering observers. Thanks to Sergio Hanquet and to Antonio Portales for their photographs of Gervais’ and probable True’s beaked whales, respectively, observed off the Canary Islands. Thanks to Colin MacLeod for his kind permission to use the distribution map of True’s beaked whale from the Digital Beaked Whale Atlas of GIS in Ecology. Also, thanks to T Sneddon and D Steel for help in the laboratory and to A van Helden, T Pusser, M Arbelo, B Brederlau, E Pérez-Gil, J Mead and R Pitman for helpful comments on the reports of this paper. We are grateful to researchers in Madeira (Filipe Alves, Luis Freitas) and Cape Verde (Vanda Marques Monteiro, Evandro Lopes and Cornelis Hazevoet) for their information on True’s beaked whales from these archipelagos. Thanks also to the academic editor of this article J Reimer, and to K Thompson and an anonymous reviewer for their positive comments which contributed to improve this paper.

Additional Information and Declarations

Competing Interests

Author Contributions

Animal Ethics

Field Study Permissions

DNA Deposition

Data Availability

The authors declare there are no competing interests.

Natacha Aguilar de Soto conceived and designed the experiments, performed the experiments, analyzed the data, contributed reagents/materials/analysis tools, wrote the paper, prepared figures and/or tables, reviewed drafts of the paper.

Vidal Martín performed the experiments, analyzed the data, contributed reagents/materials/analysis tools, prepared figures and/or tables, reviewed drafts of the paper.

Monica Silva, Roland Edler, Lisa Steiner, Michael Scheer, Roland Gockel, Petra Szlama, Ida K. Eriksson, Marisa Tejedor, Monica Perez-Gil and João Quaresma performed the experiments, contributed reagents/materials/analysis tools, reviewed drafts of the paper.

Cristel Reyes performed the experiments, analyzed the data, contributed reagents/materials/analysis tools, reviewed drafts of the paper.

Manuel Carrillo, Agustina Schiavi and Enrico Villa reviewed drafts of the paper.

Talia Morales, Belen García-Ovide, Anna Sanchez-Mora and Nerea Garcia-Tavero performed the experiments, reviewed drafts of the paper.

Dylan Walker and Wojtek Bachara contributed reagents/materials/analysis tools, reviewed drafts of the paper.

Emma Carroll conceived and designed the experiments, performed the experiments, analyzed the data, contributed reagents/materials/analysis tools, wrote the paper, reviewed drafts of the paper.

The following information was supplied relating to ethical approvals (i.e., approving body and any reference numbers):

Data were gathered from stranded whales with permits from the governments of the Azores and of El Hierro (Canary Islands). Authorization numbers were Permit #4/CN/2004, issued by the Environment Directorate of the Azores and Authorization 7021/12_dic_2012, issued by the Cabildo of El Hierro. The sighting of SECAC was performed with a permit from the Spanish Ministry MAGRAMA number 659 125 MAOT/15202. Further data are sightings in the wild collected from permitted platforms of opportunity (whale watching boats and ferries). These did not require institutional research permits for scientific data collection.

The following information was supplied relating to field study approvals (i.e., approving body and any reference numbers):

Data were gathered from stranded whales with permits from the governments of the Azores (permit number 4/CN/2004, issued 259 by the Environment Directorate of the Azores) and the Canary Islands (permit number 258 7021/12_dic_2012).

The following information was supplied regarding the deposition of DNA sequences:

The True’s beaked whale sequences described here are accessible via GenBank accession numbers: Azores CytB: KX375801, KX375802; Azores Dlp: KX150446; El Hierro CytB: KX375803; and El Hierro Dlp KX150445.

The following information was supplied regarding data availability:

The raw data are the photographs and videos included as figures in the manuscript and in the Supplemental Information, and the genetic data was uploaded to GenBank (see Data Deposition).

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
