# Peer review of "True’s beaked whale (Mesoplodon mirus) in Macaronesia"

_PeerJ, doi:10.7717/peerj.3059_

## Round 0.1 · original submission · Minor Revisions

Two reviewers have returned their comments to me. In short, both were very positive about your manuscript, and feel it is an important contribution to the literature on beaked whales. Both have, however, also added many comments and areas that need your attention and/or improvement. None of these comments are 'major' in nature, and I have found the comments to all be helpful. Therefore, my decision is 'minor revision'.

·

Basic reporting

Manuscript ID: 2016:02:9130:0:1
Title: True’s beaked whale (Mesoplodon mirus) in Macaronesia.

This manuscript describes a number of records of True’s beaked whales in the Macaronesian archipelago, as validated by standard mtDNA barcoding species identification. In addition, there are also descriptions of external morphology, indicating variation in phenotype, and defining differences between True’s and Gervais’ beaked whale that are helpful in species identification. The manuscript gives a brief overview of the distribution of True’s beaked whale records globally.

Records, of live or dead, True’s beaked whales are extremely rare and this publication makes a significant contribution to published literature on the global distributions, behavior and morphology of the species. Also, the figures and video will help enable others to better identify True’s beaked whales in the future.


General comments:

1) The manuscript would benefit from some minor editing in terms of author affiliations, changes in sentence construction, a number of typos and formatting of references.

2) The text in the discussion is rather long and would benefit from being condensed a little.

3) Ideally the manuscript would include two useful figures. Firstly, one figure giving the location of records in Macaronesia within the context of the broader global distribution of the species. Secondly, a figure giving a drawing / description of True’s showing the variation in the external morphology (and potential phenotypes), in comparison with Gervais’ beaked whale, would be really helpful to the reader.

4) Table 1, giving sightings could perhaps be a table of all records (including strandings) and placed elsewhere in the text. This could include a column to show whether the identification was validated through genetics, or through observation. Also, a column for the depth at first encounter would be interesting.

Specific comments (there are annotations on the pdf, so please check these also):

1) Author affiliations need to be standardised according to journal requirements. Some do not show the authors country.

2) Line 26: Is ‘speciose’ relevant here?

3) Line 34: Measurements should be given in SI units (metres).

4) Line 46: Suggest replacing keyword ‘genetic’ with ‘distribution’, as mtDNA and Cyt b cover genetics?

5) Line 49: Should read ‘distributions’.

6) Line 54: Should read ‘This speciose family also shows..’

7) Line 55: Remove ‘IN colour patterns’.

8) Line 67: Might read better if this sentence is broken up, i.e. ‘…species level. The position of the teeth along the bottom jaw in males….defining characteristics in both sexes.’

9) Line 68: Replace ‘very’ with ‘relatively’.

10) Line 72: Break the sentence up starting ‘True’s and Gervais’…’ Suggestions are given in the annotated pdf.

11) Line 75: Replace ‘definite’ with ‘definitive’.

12) Line 80: Constantine et al needs to be in the next sentence as this is not part of the review by MacLeod et al. 2006 OR remove the reference to MacLeod et al 2006. There is also a paper by McCann and Talbot from 1964 on a South African stranding or True's in 1959 (Proc Linn Soc Lond).

13) Line 88: Suggest replacing colon with ‘and it has been proposed..’ No ‘also’.

14) Line 91 and 92: Suggest using SI units for length’.

15) Line 96: Suggest removing ‘IN THE Azores’.

16) Line 97: Suggest replacing ‘using’ to ‘from’.

17) Line 106: Suggest amending text to ‘Observers at the beach reported that the animal may have live-stranded’.

18) Line 108: Suggest replacing comma with ‘AND no teeth were present…’ Another comment: Does this mean that no teeth had erupted, or that none were present? This animal was sexed so was confirmed to be a male?

19) Lines 113 and 115: Sambrook et al and Gilson et al are not in the reference list.

20) Line 120: Should read ‘some time’.

21) Line 123: Format for citing Geneious needs to be amended to Kearse et al., see notes on pdf.

22) Line 123: Sentence beginning ‘Species..’ should read ‘IDENTIFICATIONS WERE made’.

23) Line 140: Suggest amending sentences to, ‘any photograph, making species identification difficult. The poor quality of most photographs also make identification a challenge.’

24) Line 159: Suggest amending to ‘(accessed May and November 2015 respectively).’

25) Line 172: It would be interesting to know under what conditions the observations were made. These were opportunistic sightings covering what areas, by what type of observers, and at what type of effort level (roughly how many trips, days or hours)? This would give the reader an idea of how rare these sighting are. This would be a good place to include the sentence from Line 194, see below. Also would suggest removing ‘in’ between Sighting 1 and Table 1 and replacing it with a comma.

26) Line 194: Sentence beginning ‘All sightings..’ This is an important sentence but hangs a little here on its own. Is there anywhere else that it can be placed, perhaps a little earlier?

27) Line 213: Suggest replacing ‘quite’ with ‘relatively’. Is it possible to be more specific? Do you know how fresh?

28) Line 228: Which findings?

29) Line 229: What colouration, it would be good to be specific?

30) Line 235: Sentence beginning ‘This was observed..’ would benefit from being redrafted. Please see suggestions as annotated on the pdf.

31) Line 241: Sentence beginning ‘This similar..’ suggest changing to ‘The similarity in..’

32) Line 243: Suggest a paragraph break here? (This break may in the original text but not be so obvious in the pdf view.)

33) Line 254: Amend ‘slopping’ to ‘sloping’.

34) Line 257: Remove ‘among’.

35) Line 263: Suggest moving references to end of line to make the sentence more easy to read.

36) Line 269: Suggest amending ‘it is shared’ to ‘also’.

37) Line 270: Sentence beginning ‘Given the difficulties..’ It is not clear what is meant by this sentence and it would benefit from being redrafted for clarification.

38) Line 284: Suggest amending ‘..sightings may be a misidentification of..’ to ‘..animals may be..’

39) Lines 289-290: It would be useful to know what is meant be ‘relatively deep near shore waters’. Is there a main area that is surveyed that the authors could give a depth range for? This can be linked to Comment 25.

40) Line 293: Remove second reference to Martin 2011.

41) Line 302: Suggest amending the sentence beginning ‘The result of the..’ to ‘The preliminary genetic analyses suggests the potential for genetic population structure..’ or something similar.

42) Line 305: The final sentence hangs at the end. Perhaps another sentence would be useful? Something like: ‘Given adequate sampling, a global analyses of connectivity would prove useful in understanding gene flow between the seemingly disparate locations of sightings and strandings.’?

43) Line 314: Change ‘to’ to ‘and’ as per the annotation on pdf.

44) Line 323: ZFY and ZXF are generally in capitals.

45) Line 325: Is this a thesis?

46) Line 342: Add other authors of this paper.

47) Line 347: Add date of publication – 1994?

48) Line 349: Publisher?

49) Line 354: Remove ‘=’.

50) Line 358: Remove ‘others’ and add et al. according to journal guidelines.

51) Line 365: There is now a second edition of this text (2009).

52) General comment for figure legends: It would be useful for the reader to know the affiliations for each of the photographers in the image credits. Some of these photographers are authors but it is still useful to have the legends as standalone text.

Experimental design

No Comments.

Validity of the findings

No Comments.

Additional comments

This is a very useful summary of records and phenotypic variation in True's beaked whale records. Publishing the images, and video, will help other identify this species in the future. I hope you find the review comments useful.

Reviewer 2 ·

Basic reporting

-- Clear, unambiguous, professional English language used throughout for the most part (see minor comments under general comments to authors).
-- Literature well referenced & relevant for the most part (could do with more references to the molecular ID of beaked whale species e.g. Thompson et al., Dalebout et al.)
-- Figures are relevant, well labelled & described. Not all photographs are high resolution, but this should not be considered to detract from their relevance, because of the cryptic nature of the species under discussion
-- Most raw data supplied, but mtDNA sequences for both samples should be archived on Genbank and accession numbers provided in the manuscript.

Experimental design

No comments

Validity of the findings

-- Data is robust. Due to the opportunistic nature of beaked whale sightings, I don’t think that the criteria of “statistically sound, & controlled” are appropriate to assess here.
-- Conclusion mostly well stated, linked to original research question & limited to supporting results (see minor comments under general comments to the authors).

Additional comments

Aguilar de Soto et al. do an excellent job of advancing our knowledge of the True’s beaked whale, particularly around Macaronesia, and are appropriately cautious interpreting the somewhat sparse data available. I have some minor comments to improve the readability/clarity of the manuscript, and also some comments pertaining to the discussion relating to the authors ‘making more’ of their findings. Other than that, I think this would be a good contribution to PeerJ.

Affiliations:
Standardize to include city and country, or just country for all affiliations.

Abstract:
Minor comments:
Line 27-29: Either ‘Northern’ Hemisphere or ‘South’ Hemisphere, but don’t mix the two forms in a single sentence (I prefer ‘Northern’ and ‘Southern’ myself, but that is a personal preference)
Line 29: From your comment about the Azores/Canary Islands being the southern limit in the North Atlantic in Line 32, it seems that this distribution is assumed to be anti-tropical in the North Atlantic and Southern Hemisphere (e.g. they are not found in tropical waters). I think this is worth stating explicitly here.
Line 34-35: does this male represent a presumed adult, or subadult as well as the 2004 sample? Also, strongly suggest presenting lengths in metres not centimetres (this comment holds for whenever you mention lengths throughout the entire manuscript)
Line 34: You describe the findings from the stranded animals in adequate detail, but don’t mention any findings based off the underwater video. Did this allow you to describe coloration patterns as well? Anything else of interest? At the very least, does this suggest that the Azores could be a location for behavioral studies of True’s similar to the work that Whitehead et al. have done on bottlenosed whales off the Gully area?
Line 41: colourations should just be ‘colouration’. Sightings can probably just be ‘sighting’.
Line 41-43: This seems like a bit of a ‘whelming’ way to end your abstract. I’d suggest replacing these lines with a conclusion highlighting your findings: e.g. describing a new coloration pattern from animals confirmed to be True’s beaked whales based on molecular ID (i.e. these new coloration patterns cannot be attributed to a case of mistaken identity), as well as suggesting the Azores/Canary Islands could be unique locations for studying the behavior of the enigmatic True’s beaked whale, based on the fact that live video footage was collected for this species in this location.

Introduction:
Minor comments:
Line 59: genus should be ‘genera’
Line 63-64: It would be useful to give an exact context for the size of these species in parentheses e.g. Cuviers beaked whales (~5-7 m) and bottlenose whales (~9-10 m)
Line 64-65: Either refer to the species by their common names, or by their scientific names, but don’t flip-flop back and forth e.g. Z. cavirostris should probably be Cuvier’s beaked whale to follow your usage in the previous and following sentences.
Line 66: Suggest adding in approximate size for mesoplodonts as well e.g. Mesoplodonts are similar in size to each other (~4-6 m)
Lines 68-70: Are these statements covered by the MacLeod et al. (2006) reference as well?
Lines 74-76: It is kind of obvious, but it might be worth explicitly stating that this cue can’t be used to distinguish the females of the species at sea.
Lines 59-78: I think this paragraph would be aided by a cartoon figure summarizing the key differences in morphology between the species mentioned. I know making these types of figures can be a pain but it would literally just involve tracing the head shape of some of the photos used in the figures of the manuscript.
Lines 88-90: I don’t think this relevant to the current manuscript, because you do not report any sightings of True’s beaked whales from Cape Verde.
Line 93: Replace M. mirus with True’s beaked whale (i.e. refer to it as its common name because you do this almost everywhere else in the manuscript).
Line 92-94: Was this presumably an adult male? Worth explicitly mentioning if you consider it an adult, given you clarified the previous specimen was sub-adult.
Line 96: I think you need more of a segue into this final paragraph of your introduction. You’ve set up the fact that it can be difficult to diagnose between Mesoplodon species based on morphology… I think you need a sentence basically saying that one way you can reliably distinguish between ziphiid species is the use of molecular markers. There are a number of papers by Dalebout et al. and Thompson et al. on the use of molecular markers for identifying ziphiid species that could be cited in support of this.

Methods
Minor comments:
Line 119: ‘using standard protocols’ is a little vague. Did you follow the methods of Dalebout et al. for these PCR reactions? It would be more specific to mention that if so.
Line 122: Presumably BigDye v3.1 terminator rather than v1.1? Also, TM should be superscript.
Line 123: Edited how? For sequencing quality? Did you use PHRED scores and/or do this by eye?
Line 124: ‘of’ in this sentence can be removed
Line 126: Presumably you carried out this comparison through BLAST. Should state this explicitly.
Line 129: A map with the location of the stranded animals and where the live sightings/underwater footage was taken would be a good addition to this manuscript.

Results
Minor comments:
Line 155-156: What are the percentages you are reporting: bootstrap scores, or sequence identity to the reference sequences?
Line 159: suggest changing ‘when accessed’ to ‘when BLAST was accessed’
Line 163: suggest changing ‘mtDNA’ to ‘mtDNA control region’
Line 166: capitalize E-value.
Line 167: suggest changing ‘a beaked whale’ to ‘a True’s beaked whale’
Line 168: suggest changing ‘a whale’ to ‘a True’s beaked whale’
Line 174: suggest ‘horizontally’ should be ‘dorsally’
Line 194: Add a reference to the figures showing the Gervais’ beaked whales

Discussion
Overall comments:
The authors provide a detailed explanation of the difficulties in distinguishing True’s beaked whales from other related species, including Gervais’ beaked whales, based on morphology, behavior and distribution. I thought, however, that the discussion omitted an important way they can be distinguished with accuracy: DNA. I think that it would be worth adding at least a few sentence about the value of molecular ID for identification of True’s beaked whales. I also think their live sightings indicate the potential for the Canary Island’s/Azores to act as a potential area for behavior studies on the True’s beaked whale. I think they could also make more of this in the discussion.

Minor comments:
Line 216: Does this suggest that some sightings recorded as Cuvier’s beaked whales might in fact be True’s beaked whales, if this coloration is more widespread than just occurring in this single individual?
Line 219: True’s beaked whale not M. mirus
Line 219-220: Is there any possibility of a change in pigmentation with age, given this individual was a subadult male? You mention ontogenetic shifts in coloration in Lines 243 - potentially it would be worth advancing this to here where you discuss the coloration of the juvenile animal?
Line 229: suggest changing the word ‘clear’ for ‘pale’ or ‘white’
Line 298: Unclear who ‘these authors’ refer to.

References:
Italics missing for species names.

Figures and Tables
I thought all figures and tables did a good job demonstrating the authors’ points.

---

## Round 0.2 · Minor Revisions

I have heard back from two reviewers, who were very happy with your revisions, and there are now only a few small things left to take care of, hence "Minor revisions" is my decision.

·

Basic reporting

No Comments

Experimental design

No Comments

Validity of the findings

No Comments

Additional comments

The manuscript has been greatly improved by the revision and now reads well.

There are a number of very minor comments that should also be addressed before final publication:

1. Line 89: ‘later’ should read ‘latter’
2. Table 1 is really useful, suggest using ‘No.’ for Number of animals, rather than no, also might be good to clarify this in the legend.
3. Line 313: The sentence beginning ‘The two sightings..’ might read better as ‘The two sightings classified as ‘possible’ True’s beaked whales in the Canary Islands were both observed to breach repeatedly (Figures 10, 11).’ Or something like this for clarification.
4. Some inconsistencies in the references. Some have DOI’s some do not and some author initials have full stops some do not. See lines: 502, 505, 510, 533, 537-538, 544 highlighted on the annotated pdf attached.

All the best to the authors on this interesting publication.

Reviewer 2 ·

Basic reporting

No comments

Experimental design

No comments

Validity of the findings

No comments

Additional comments

I am very satisfied with the way the authors responded to the comments from both the reviewers, and I will be excited to see this manuscript in print. Everything is reading really well, and I enjoyed getting to go through this paper again. I believe it hits all the requirements of PeerJ in terms of basic reporting, experimental design and validity of the findings. No need to include an answer to this query in the manuscript (just my own musing), but do you think the apparent absence from the North Pacific is real or an artifact of sampling? Even though the True’s don’t share this distribution with any of the other beaked whales, it is the same distribution as long-finned pilot whales, which are also absent from the North Pacific, and are another squid-feeding species…I do have a few last-minute very, very minor suggestions below.

Affiliations: At line 17 suggest changing the postcode to ‘England’, seeing as your Scottish co-authors have Scotland listed and not a postcode.

Abstract:
Line 27: I think ‘poorly studied’ should potentially be ‘poorly known to science’ or some other equivalent (there is some ambiguity here because this could be a judgment call saying previous research was “poor”)
Line 38: whale and that should be reversed in order

Introduction:
Line 55: Ziphiidae shouldn’t be capitalized (only genus and species names, not family names)
Line 58: suggest changing ‘deep waters’ to ‘depth’
Line 73: Suggest starting this sentence with ‘In contrast,’ (to explicitly contrast how tricky the mesoplodonts are in comparison with the Cuvier’s and bottlenose whale IDs)
Line 78: Mesoplodont shouldn’t be italicized because it is the ‘common’ name equivalent of genus Mesoploodon
Line 79-82: I feel like you might need to drop a reference in here for these sentences
Line 122: suggest changing ‘close’ to ‘close-up’

Methods:
Line 143: Suggest changing ‘using standard protocols’ to ‘following standard protocols’, because you’ve already used using a lot in that sentence. Then you could ditch the ‘following’ in (following Dalebout).
Line 152: Reference for BLAST:
Altschul, S. F., W. Gish, W. Miller, E. W. Myers, and D. J. Lipman. 1990. Basic local alignment search tool. J. Mol. Biol. 215:403–410.
Line 153: missing ‘m’ after 3.5
Line 180: this sentence reads like results, not methods. Suggest moving this to results and starting with the sentence on Line 181 (where you can refer to Table 1 parenthetically). Actually, much of the rest of the methods (Lines 186 onwards) are results, not methods. Suggest moving all of this to results section.

Results:
Line 311: I believe the report numbers should be 5, 9 and 11, not 5, 6, 9
Line 314: ‘on’ should be ‘in’
Line 315: ‘on’ should be ‘in’

Discussion:
Line 343: I don’t know if ‘contrasting’ is really the word you want here. Maybe differing?
Line 357: ‘were’ instead of ‘are’
Line 382: ‘was’ instead of ‘were’
Line 389: unclear what ‘slightly upwards in True’s’ refers to. The edge of the mouthline? State this explicitly
Line 391: add ‘(also not always present)’ – you mentioned the pale band was not always present in True’s.
Line 410: Ziphiids should not be capitalized
Line 433: Based on the coordinates given, report 2 in Table 1 is actually closest to the coast. You could amend this sentence to refer to it being the sighting that occurred in the shallowest water, which is still accurate.

References:
In general: sometimes DOIs have been included for references, sometimes not. Am guessing this will need to be standardized.
Line 510: Need to remove Clarige DE etc etc from the end of this reference.
Lines 530, 597: Remove “Scott” and change to Baker, CS
Lines 514, 531, 576, 606: Italicize species names
Line 547: remove parentheses on date. Italicize journal name.
Line 558: Standardize capitalization for this reference
Line 593: irregular capitalization on Ferdana
Line 604: Capitalize Odontoceti

Figures and Tables
I thought Table 1, Figure 14 (lovely illustration!), and Figure 16 were a great addition to the manuscript

Congratulations to the authors on a great manuscript!

---

## Round 0.3 · accepted · Accept

The minor revisions are satisfactory, and this paper is ready to be published. I look forward to seeing your paper online!